# How to GAN event subtraction

**Anja Butter, Tilman Plehn and Ramon Winterhalder***

Institut für Theoretische Physik, Universität Heidelberg, Germany

* winterhalder@thphys.uni-heidelberg.de

## Abstract

Subtracting event samples is a common task in LHC simulation and analysis, and standard solutions tend to be inefficient. We employ generative adversarial networks to produce new event samples with a phase space distribution corresponding to added or subtracted input samples. We first illustrate for a toy example how such a network beats the statistical limitations of the training data. We then show how such a network can be used to subtract background events or to include non-local collinear subtraction events at the level of unweighted 4-vector events.

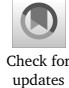
## Content

# 1 Introduction

Modern analyses of LHC data are increasingly based on a data-to-data comparison of measured and simulated events. The theoretical basis of this approach are generated samples of unweighted or weighted LHC events. To match the experimental precision such samples have to be generated beyond leading order in QCD. In modern approaches to perturbative QCD at the LHC such simulations include subtraction terms, leading to events with negative weights. Examples for such subtraction event samples are subtraction terms for fixed-order real emission [1–5], multi-jet merging including a parton shower [6,7], on-shell subtraction [8], or the subtraction of precisely known backgrounds [9].

Generative adversarial networks or GANs [10] are neural networks which naturally lend themselves to operations on event samples, as we will show in this paper. Such generative networks have been proposed for a wide range of tasks related to LHC event simulation and are expected to lead to significant progress once they become part of the standard tool box. This includes for instance phase space integration [11], event generation [12–15], detector simulations [16–22], unfolding [23], parton showers [24–28], or searches for physics beyond the Standard Model [29]. Most recently, we have shown that fully conditional GANs can be used to invert typical Monte Carlo processes at the LHC, like for instance a fast detector simulation [30].

In this paper, we show how GANs can perform simple operations on event samples, namely adding and subtracting existing samples. Such a network is trained to generate unweighted events with a phase space density corresponding to a sum or difference of two or more input samples. We will illustrate the idea behind a generative event sample subtraction and addition in Sec. 2. This example shows how generative networks can beat the statistical limitations of the training samples. Specifically, we produce events with statistical fluctuations which are significantly smaller than the corresponding statistical fluctuations of the training data. The feature behind this naively impossible improvement are the excellent interpolation properties of neural networks in a high-dimensional phase space.

In Sec. 3 we will then subtract unweighted 4-vector events for the LHC in two examples. First, we subtract the photon continuum from the complete Drell–Yan process and find the $Z$-pole and the known interference patterns. This can be seen as a toy example for a background subtraction at the level of parton-level event samples. For instance, this setup could allow us to study the kinematics of four-body decay signals, simulated to high precision from observed background and signal-plus-background samples.

Finally, we combine a hard matrix element for jet radiation with collinear subtraction events. This gives us an event sample that follows the matrix element minus the subtraction term without any intermediate binning in the phase space. We show how this subtraction works even if we do not make use of the local structure of the subtraction terms. It illustrates how simulations in perturbative QCD might benefit from GANs, in soft-collinear subtraction, on-shell subtraction, or a veto-like combination of phase space and parton shower.

# 2 Toy example

The advantage of GANs learning how to subtract event samples can be seen easily from statistical uncertainties in event counts. Traditionally, we generate the two samples and combine them through some kind of histogram. If we start with $N + n$ events and subtract $N \gg n$

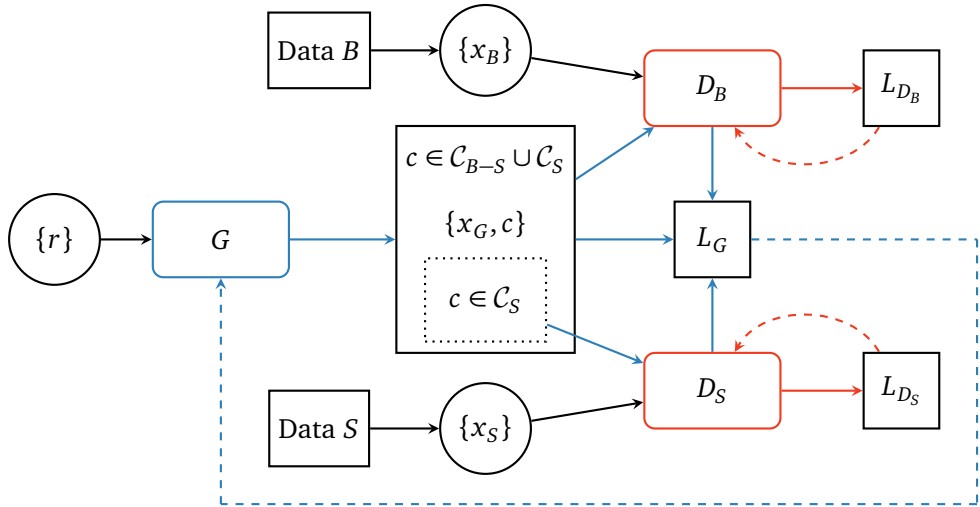

Figure 1: Structure of our subtraction GAN. The input $\{r\}$ describes a batch of random numbers and $\{x_{B,S}\}$ the true input data batches. The label $c$ encodes the category of the generated events. Blue arrows indicate the generator training, red arrows the discriminators training.

statistically independent events, the uncertainty on the combined events in one bin is given by

$$\Delta_n = \sqrt{\Delta_{N+n}^2 + \Delta_N^2} \approx \sqrt{2N} \gg \sqrt{n} \ . \tag{1}$$

In any bin-wise analysis the bin width has to be optimized. On the one hand larger bins with more events per bin minimize the relative statistical error, but on the other hand they reduce the resolution of features.

In our GAN approach we avoid defining such histograms and replace the explicit event subtraction by a subtraction of interpolated sample properties over phase space. We will first develop this approach in terms of a simple toy example and then show how it can be extended to unweighted 4-vector events as used in LHC simulations. Unfortunately, there does not (yet) exist a rigid description of statistical and systematic uncertainties associated with GANs, but we will show how the fluctuations we observe in our generated samples are visibly smaller than what we would expect from the input data and Eq.(1).

## 2.1 Single subtraction

We start with a simple 1-dimensional toy model, *i.e.* toy events which are described by a single real number $x$. We then define a base distribution $P_B$ and a subtraction distribution $P_S$ as

$$P_B(x) = \frac{1}{x} + 0.1 \qquad \text{and} \qquad P_S(x) = \frac{1}{x} \ . \tag{2}$$

The target distribution for the subtraction is then

$$P_{B-S} = 0.1 \ . \tag{3}$$

To produce unweighted subtracted events our GAN is trained to generate the event sets $\{x_B\}$ and $\{x_S\}$ simultaneously. It thereby learns the distribution $P_{B-S}$ using the information encoded in the two input samples.

The corresponding GAN architecture is shown in Fig. 1 and consists of a generator and two independent discriminators, one for each dataset. The generator takes random noise $\{r\}$

as input and generates samples $\{x_G, c\}$, where $x_G$ stands for an event and $c$ for a label. The underlying idea is to start from an event sample which follows $P_B$ and split it into two mutually exclusive samples following $P_S$ and $P_{B-S}$, with class labels $\mathcal{C}_S$ or $\mathcal{C}_{B-S}$. During training we demand that the distribution over events from class $\mathcal{C}_S$ follow $P_S$ while the full event sample follows $P_B$. After normalizing all samples correctly the events with class label $\mathcal{C}_{B-S}$ will then follow the distribution $P_{B-S}$.

Technically, the class label $c$ attached to each event is a real 2-dimensional vector, such that it can be manipulated by the network. Through a SoftMax function in the final generator layer the entries of $c$ are forced into the interval $[0, 1]$ and sum to 1. We then create a so-called one-hot encoding by mapping $c$ to

$$c_i^{\text{one-hot}} = \begin{cases} 1 & \text{if } c_i = \max(c) \\ 0 & \text{else }. \end{cases} \tag{4}$$

This representation is two-dimensional binary and most convenient for manipulating the samples. We can use it to define the label classes via $\mathcal{C}_i = \{c \mid c_i^{\text{one-hot}} = 1\}$.

In Fig. 1 we see that for the class $\mathcal{C}_S$ and the union of $\mathcal{C}_S$ with $\mathcal{C}_{B-S}$ we train the discriminators to distinguish between events from the input samples and the generated events. The training of the discriminators $D_i$ corresponding to the two input samples $\{x_S\}$ and $\{x_B\}$ uses the standard discriminator loss function for instance in the conventions of Ref. [15]

$$L_{D_i} = \left\langle -\log D_i(x) \right\rangle_{x \sim P_T} + \left\langle -\log(1 - D_i(x)) \right\rangle_{x \sim P_G}. \tag{5}$$

We add a regularization and obtain the regularized Jensen-Shannon loss function

$$L_{D_i}^{(\text{reg})} = L_{D_i} + \lambda_{D_i} \left\langle (1 - D_i(x))^2 \, |\nabla \phi_i|^2 \right\rangle_{x \sim P_T} + \lambda_{D_i} \left\langle D_i(x)^2 \, |\nabla \phi_i|^2 \right\rangle_{x \sim P_G}, \tag{6}$$

where we define

$$\phi_i(x) = \log \frac{D_i(x)}{1 - D_i(x)}. \tag{7}$$

In parallel, we train the generator to fool the discriminators by minimizing

$$L_G = \sum_i \left\langle -\log D_i(x) \right\rangle_{x \sim P_G}. \tag{8}$$

An additional aspect in manipulating samples is that we need to keep track of the normalization or number of events in each class. To generate a clear and differentiable assignment we introduce the function

$$f(c) = e^{-\alpha(\max(c)^2 - 1)^{2\beta}} \in [0, 1] \qquad \text{for} \qquad 0 \leq c_i \leq 1. \tag{9}$$

Adapting $\alpha$ and $\beta$ we can make the gradient around the maximum steeper and push $f(0) \to 0$. In that case $f(c) \approx 1$ only if one of the entries of $c_i \approx 1$. By adding

$$L_G^{(\text{class})} = \left(1 - \frac{1}{b} \sum_{c \in batch} f(c)\right)^2 \tag{10}$$

to the loss function we reward a clear assignment of each event to one class and generate a clear separation between classes. Finally, we use the counting function in combination with masking to fix the normalization of each sample with

$$L_{G_i}^{(\text{norm})} = \left(\frac{\sum_{c \in \mathcal{C}_i} f(c)}{\sum_{c \in \mathcal{C}_B} f(c)} - \frac{\sigma_i}{\sigma_0}\right)^2. \tag{11}$$

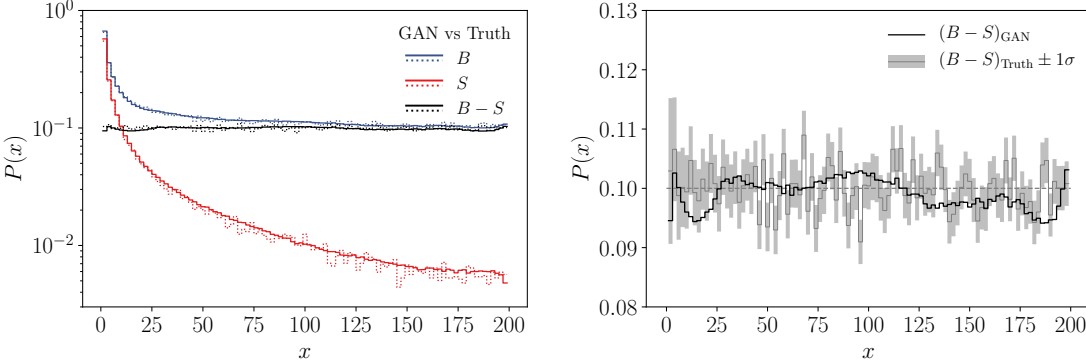

Figure 2: Left: Generated (solid) and true (dashed) events for the two input distributions and the subtracted output. Right: distribution of the subtracted events, true and generated, including the error envelope propagated from the input statistics.

Adding these losses to the generator loss we get

$$L_G \rightarrow L_G^{(\text{full})} = L_G + \lambda_{\text{class}} L_G^{(\text{class})} + \lambda_{\text{norm}} L_G^{(\text{norm})} , \qquad (12)$$

with properly chosen factors $\lambda_{\text{class}}$ and $\lambda_{\text{norm}}$. In this paper we always use $\lambda_{\text{class}} = \lambda_{\text{norm}} = 1$. For the denominator in Eq.(11) we always choose the approximation of the number of predicted events in the base class $\mathcal{C}_B = \mathcal{C}_{B-S} \cup \mathcal{C}_S$ as reference value. The integrated rates $\sigma_i$ have to be given externally. For our toy model we can compute them analytically while for an LHC application they are given by the cross section from the Monte Carlo simulation.

Our GAN uses a vector of random numbers as input. The size of the vector has to be at least the number of degrees of freedom. For the implementation we use KERAS 2.2.4 [31] with a TENSORFLOW 1.14 back-end [32]. The discriminator and generator networks consist of 5 layers with 128 units per layer using the ELU activation function. With $\lambda_{D_i} = 5 \cdot 10^{-5}$ and a batch size of 1024 events, we run for 4000 epochs. Each epoch consists of one update of the generator and 20 updates of the discriminator. We found that the intense training of the discriminator is necessary to reach sufficiently precise results. To obtain a good separation of the classes with $f(c)$ we set $\alpha = 10$ and $\beta = 1$. Finally, using the ADAM [33] optimizer throughout this paper, we choose a learning rate of $3 \cdot 10^{-4}$ for generator and discriminator and a large decay of the learning rate of $2 \cdot 10^{-2}$ for the discriminator which stabilizes the training. The decay for the generator is slightly smaller with $5 \cdot 10^{-3}$. Our training datasets consist of $10^5$ samples for each dataset $\{x_S\}$ and $\{x_B\}$.

We show numerical results for a single GAN subtraction and analyze the size of the statistical fluctuations in Fig. 2. In the left panel we show the two input distributions defined in Eq.(2), as well as the true and generated subtracted distribution. The dotted lines illustrate the shape of the training dataset, while the full lines show the generated distribution using $5 \cdot 10^6$ events. The former two distributions only serve to confirm that the GAN learns the input information correctly. The generated subtracted events indeed follow the probability distribution in Eq.(3). Aside from the fact that all three distributions show excellent agreement between truth and GANned events, we see how the neural network interpolates especially in the tail of the distribution. In the right panel of Fig. 2 we zoom into the subtracted sample to compare the statistical uncertainties from the input data with the behavior of the GAN. The uncertainty is estimated from the number of events per bin in the base and subtraction histogram $N_B$ and $N_S$, taking into account the corresponding normalization factors $n_B$ and $n_S$. In analogy to

Eq.(1) we compute it as

$$
\begin{aligned}
\Delta_{B-S} &= \Delta_{n_B N_B - n_S N_S} \\
&= \sqrt{\Delta_{n_B N_B}^2 + \Delta_{n_S N_S}^2} \\
&= \sqrt{n_B^2 N_B + n_S^2 N_S} \, .
\end{aligned}
\tag{13}
$$

As mentioned above, we expect the GAN to deliver more stable results than we could expect from the input sample, because the GAN interpolates all input distributions. This way we avoid a bin-by-bin statistical uncertainty of the subtracted sample. Indeed, our subtracted curve in the right panel of Fig. 2 lies safely within the $1\sigma$ region of the data. The statistical fluctuations of the GANned events are much smaller than the statistical fluctuations in the input data. On the other hand, the GANned distribution shows systematic deviations, but also at a visibly smaller level than the statistical fluctuation of the input data. While this observation does not imply a proof that GANs can beat the statistical limitations of the input data, they give a clear hint that the interpolation properties can balance statistics at some level.

## 2.2 Combined subtraction and addition

To show how our approach could be generalized to subtracting and adding any number of event samples we can extend our single subtraction toy model by a third sample to be added to the difference described in Eq.(3). We now consider three samples corresponding to the 1-dimensional distributions

$$
P_B(x) = \frac{1}{x} + 0.1 \qquad P_S(x) = \frac{1}{x} \qquad P_A(x) = \frac{m}{\pi} \frac{\gamma}{\gamma^2 + (x - x_0)^2} \, .
\tag{14}
$$

As a third input we add the Breit-Wigner distribution $P_A$, so our target distribution becomes

$$
\begin{aligned}
P_{B-S+A} &= \frac{m}{\pi} \frac{\gamma}{\gamma^2 + (x - x_0)^2} + 0.1 \\
&= \frac{5}{\pi} \frac{10}{100 + (x - 90)^2} + 0.1 \, ,
\end{aligned}
\tag{15}
$$

for the values $m = 5$, $\gamma = 10$, and $x_0 = 90$. We now sample $\{x_B\}$, $\{x_S\}$ and $\{x_A\}$ individually from the input distributions and want to learn the probability distribution $P_{B-S+A}$. The approach is the same as described before, but for three classes as shown in Tab. 1 and a three-dimensional class vector. Treating the subtraction exactly as before we obtain our target distribution $P_{B-S+A}$ by adding the event with class $\mathcal{C}_A$.

Compared to the sample subtraction introduced before, adding samples is obviously not a big challenge. In principle, we could just add the unweighted event samples in the correct

Table 1: Category assignment for a combined addition and subtraction of three samples.

|  | $\mathcal{C}_{B-S}$ | $\mathcal{C}_S$ | $\mathcal{C}_A$ |
|---|---|---|---|
| Data $B$ | 1 | 1 | 0 |
| Data $S$ | 0 | 1 | 0 |
| Data $A$ | 0 | 0 | 1 |
| $B-S+A$ | 1 | 0 | 1 |

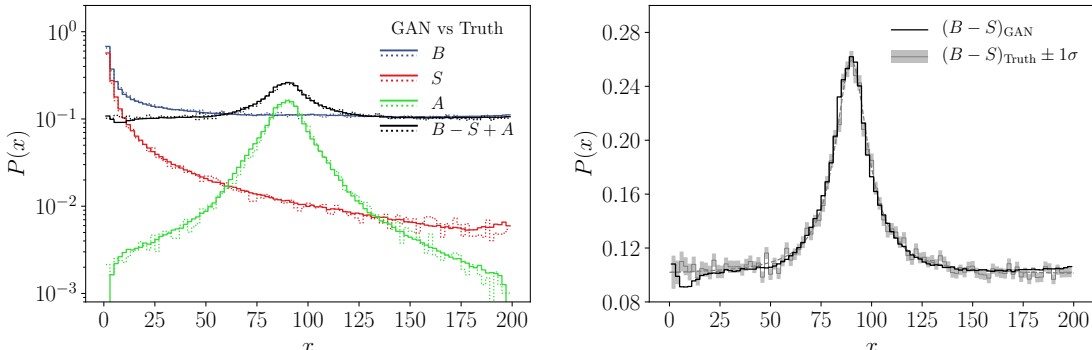

Figure 3: Left: Generated (solid) and true (dotted) events for the three input distributions and the combined output. Right: distribution of the combined events, true and generated, including the error envelope propagated from the input statistics.

proportion, learn the phase space structure with a GAN, and then generate any number of events very efficiently. The reason why we discuss this aspect here is that it shows how our subtraction GAN can be generalized easily.

In Fig. 3 we show the numerical results of subtracting one distribution $\{x_S\}$ from the base distribution $\{x_B\}$ and adding a second distribution $\{x_A\}$ with a distinct feature. As before, this combination is learned from the three input distributions without binning the corresponding phase space. The hyper-parameters are slightly modified with respect to the simple subtraction model. The networks now consist of 7 layers with 128 units which we train for 1000 epochs with 4 iterations. We fix the relative weight of the gradient penalty to $\lambda_{D_i} = 5 \cdot 10^{-5}$. The separation of the three classes is efficient for $\alpha = 5$ and $\beta = 1$. Finally, we set the learning rate to $8 \cdot 10^{-4}$ and its decay to $2 \cdot 10^{-2}$ for generator and discriminator. The remaining parameters are the same as for the pure subtraction case. In the left panel of Fig. 3 we confirm that the GAN indeed learns the three input structures correctly and interpolates each of them smoothly. We also see that the generated events follow the combination $B - S + A$ with its flat tails and the central Breit–Wigner shape. As for the pure subtraction in Fig. 2 we also compare the statistical fluctuation of the binned input data with the behavior of the GANned events. The GAN extracts the additional Breit–Wigner feature with high precision, but, as always, some systematic deviations arise in the tails of the distribution.

## 2.3 General setup

Finally, we note that our network setup is not limited to three classes. We can generalize it to a base distribution, $M$ subtraction datasets, and $N$ added datasets. The corresponding category assignment, generalized from Tab.1, is given in Tab. 2 and encoded in an enlarged classification vector $c$. The base class is then defined as

$$\mathcal{C} = \bigcup_{i=0}^{M} C_i \,. \tag{16}$$

In this case the network has to learn all $M + N + 1$ input distributions through individual discriminators $D_B$, $D_{S_i}$, and $D_{A_j}$ with $i \leq M$ and $j \leq N$. The rough structure of the network is given in Fig. 4. The training of the generator follows directly from the description above. While we do not benchmark this extended setup in this paper, we expect it to be useful when a set of subtraction terms accounts for different features, and splitting them improves their simulation properties.

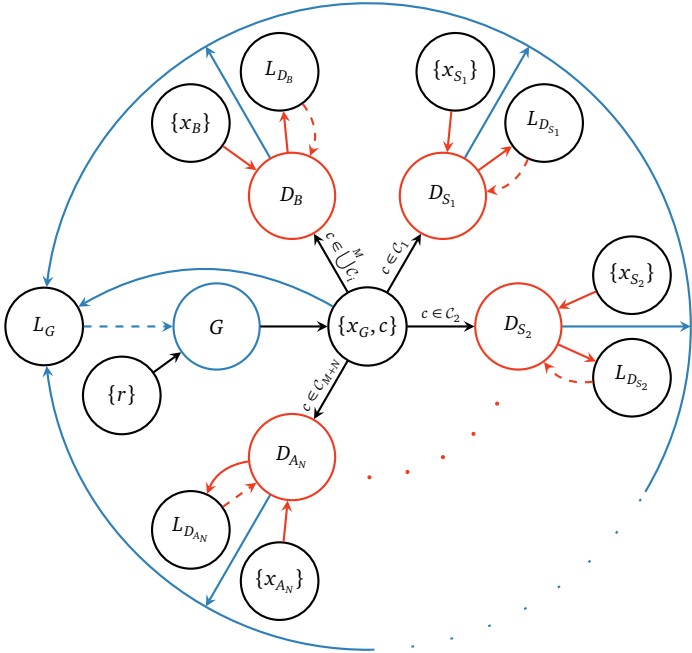

Figure 4: Structure of our general subtraction and addition GAN. The input $\{r\}$ describes a batch of random numbers and $\{x\}$ the true input data or generated batches. The label $c$ encodes the category of the generated events. Blue arrows indicate the generator training, red arrows the discriminators training.

Until now we have always assumed that we can subtract a sample $\{x_S\}$ from a sample $\{x_B\}$ and find a well-behaved distribution for $B-S$. Specifically, the resulting probability $P_{B-S}$ should be positive all over phase space. This is not always the case. First, we note that a global sign of the combination is not a problem, because we can always learn $S-B$ instead of $B-S$. Next, changing signs in the $S$ or $B$ contributions can be accommodated by splitting the respective sample according to the sign and applying the combined subtraction and addition described in Sec. 2.2. A phase-space dependent sign in $S-B$ could be most easily accommodated by adding a constant off-set either by hand or again using the combined subtraction and addition. A typical example would be to add the Born term to the virtual correction before subtracting

Table 2: Details for the category selection in the general case.

|  | $\mathcal{C}_0$ | $\mathcal{C}_1$ | $\mathcal{C}_2$ | $\cdots$ | $\mathcal{C}_M$ | $\mathcal{C}_{M+1}$ | $\cdots$ | $\mathcal{C}_{M+N}$ |
|---|---|---|---|---|---|---|---|---|
| Data $B$ | 1 | 1 | 1 | $\cdots$ | 1 | 0 | $\cdots$ | 0 |
| Data $S_1$ | 0 | 1 | 0 | $\cdots$ | 0 | 0 | $\cdots$ | 0 |
| Data $S_2$ | 0 | 0 | 1 |  | 0 | 0 | $\cdots$ | 0 |
| $\vdots$ | $\vdots$ | $\vdots$ | $\vdots$ | $\ddots$ |  | $\vdots$ |  | $\vdots$ |
| Data $S_M$ | 0 | 0 | 0 |  | 1 | 0 | $\cdots$ | 0 |
| Data $A_1$ | 0 | 0 | 0 | $\cdots$ | 0 | 1 |  | 0 |
| $\vdots$ | $\vdots$ | $\vdots$ | $\vdots$ | $\ddots$ | $\vdots$ |  | $\ddots$ |  |
| Data $A_N$ | 0 | 0 | 0 | $\cdots$ | 0 | 0 |  | 1 |
| Combination | 1 | 0 | 0 | $\cdots$ | 0 | 1 | $\cdots$ | 1 |

the dipole.

In cases where this is not a suitable solution, we can replace the categories $\mathcal{C}_{B-S}$ and $\mathcal{C}_S$ by the three categories $\mathcal{C}_{B\cap S}$, $\mathcal{C}_{B\setminus S}$, and $\mathcal{C}_{S\setminus B}$. They indicate events corresponding to $B$ and $S$, only $B$, or only $S$. The discriminator compares for instance the combination of $\mathcal{C}_{B\cap S}$ and $\mathcal{C}_{B\setminus S}$ with the $B$-data. The difference $B-S$ will be given by events with label $\mathcal{C}_{B\setminus S}$ in regions where $B > S$ and events with label $\mathcal{C}_{S\setminus B}$ in regions where $B < S$, the latter weighted with weight minus one. While this simple extension of the label vector is very straightforward, the third category induces an additional degree of freedom in the way the network can distribute events into different categories. This freedom needs to be constrained to prevent the network from simply assigning for instance all events into the categories $\mathcal{C}_{B\setminus S}$, and $\mathcal{C}_{S\setminus B}$. A possible solution would be to maximize the number of events in $\mathcal{C}_{B\cap S}$ via a term in the loss function and force the network to share as many events between the distributions as possible.

# 3   LHC events

After showing how it is possible to GAN-subtract 1-dimensional event samples from each other we have to show how such a tool can be applied in LHC physics. In this case the (unweighted) events are 4-momenta of external particles. We ignore all information on the particle identification, except for its mass, which allows us to reduce external 4-momenta to external 3-momenta [15, 30]. Because the input events might have been object to detector effects we do not assume energy-momentum conservation for the entire event. This means that the network has to learn the 4-dimensional energy-momentum conservation and this subtraction of simple LHC events is inherently multi-dimensional. We will present two simple examples for LHC event subtraction, the separation of on-shell photon and $Z$ contributions to the Drell-Yan process and the subtraction of collinear gluon radiation in $Z$+jet production.

## 3.1   Background subtraction

Our first example for event subtraction at the LHC is the Drell–Yan process, which receives contributions with distinct phase space features from the photon and from the $Z$-boson, as seen in Fig. 5. The specific question in our setup is if we can subtract a background-like photon continuum contribution from the full process and generate events only for the $Z$-exchange combined with the interference term,

$$\begin{aligned} B: \quad & pp \to e^+ e^- \\ S: \quad & pp \to \gamma \to e^+ e^- \, . \end{aligned} \qquad (17)$$

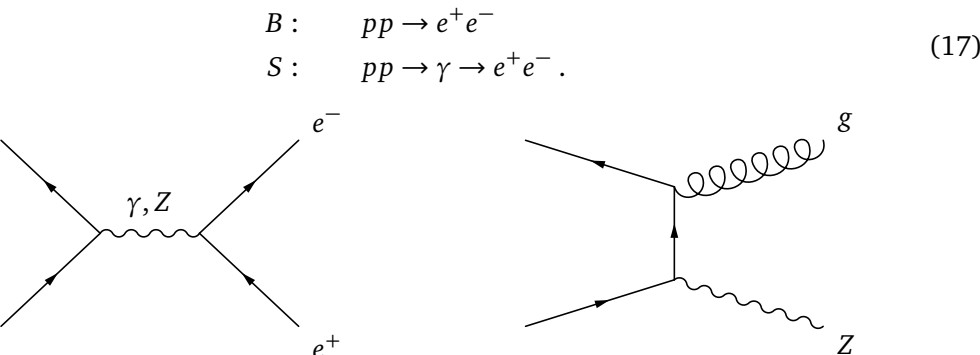

Figure 5: Sample Feynman diagrams for the background subtraction (left) and collinear subtraction (right) applications.

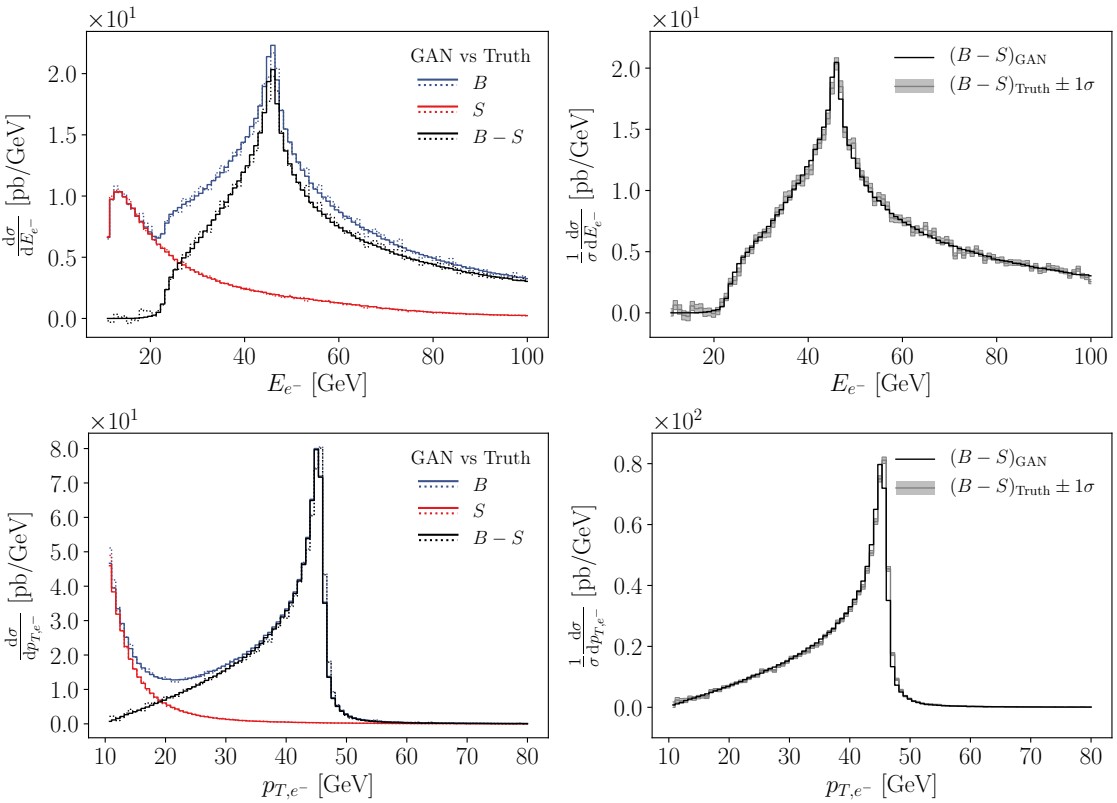

Figure 6: Left: Generated (solid) and true (dashed) $e^+e^-$ events at the LHC for the two input distributions and the subtracted output. Right: distribution of the subtracted events, true and generated, including the error envelope propagated from the input statistics.

We generate 1M events with MADGRAPH5 [34] for an LHC energy of 13 TeV, applying minimal cuts on the outgoing electrons. We require a minimal $p_T$ of 10 GeV, a maximal rapidity of 2.5 for each electron, and a minimal angular separation of 0.4. We do not apply a detector simulation at this stage, because our focus is on comparing the generated and true distributions, and we have shown that detector simulations can be included trivially in our GAN setup [15,30].

Aside from the increased dimension of the phase space the subtraction GAN has exactly the same structure as the toy example of Sec. 2. The hyper-parameters have to be adjusted to the increased dimensionality of the phase space. We use a 16-dimensional latent space. The discriminator and generator networks consist of 8 layers with 80 and 160 units per layer, respectively. In this high-dimensional case we use the LeakyRelu activation function. Further, we choose $\lambda_{D_i} = 10^{-5}$ and a batch size of 1024 events and train for 1000 epochs. Each epoch consists of 5 iterations in which the discriminator gets updated twice as much as the generator. For a proper separation of the classes with $f(c)$ we set $\alpha = 5$ and $\beta = 1$. Finally, we choose a large decay of the learning rate of $10^{-2}$ which stabilizes the training and pick a learning rate at the beginning of $10^{-3}$. Our training datasets consist of $10^5$ samples for each dataset $\{x_B\}$ and $\{x_S\}$.

In Fig. 6 we show the performance of the LHC event subtraction for two example distributions. First, we clearly see the $Z$-mass peak in the lepton energy of the full sample, compared with the feature-less photon continuum in the subtraction sample. The subtracted curve is expected to describe the $Z$-contribution and the interference. It smoothly approaches zero for small lepton energies, where the interference is negligible. Above that we see the Jacobian

peak from the on-shell decay, and for larger energies a small interference term enhancing the high-energy tail. In the (usual) right panel we show the subtracted curve including the statistical uncertainties from the input samples. As the second observable we show the transverse momentum of the electron. Here the $Z$-pole appears as a softened endpoint at $m_Z/2$. The photon continuum dominates the combined distribution for small transverse momenta. Indeed, the GAN-subtracted on-shell and interference contribution is localized around the endpoint, with a minor shift in the resolution at the edge.

Obviously, our subtraction of the background to a di-electron resonance is not a state-of-the-art problem in LHC physics. A more interesting application of our method could be four-body decays. We could start from a combined signal plus background sample of Higgs decays to four fermions, generate a background-only sample using control regions, and then GAN a set of signal events. While in a regular analysis the events we obtain from subtracting a background from the signal-plus-background sample do not reflect the signal properties, our GANned subtraction events should reflect all kinematic features of the signal events in the data.

## 3.2 Collinear subtraction

The second example for event subtraction at the LHC is collinear radiation off the initial state, for instance

$$
\begin{aligned}
B: & \quad pp \to Zg \quad \text{(matrix element)} \\
S: & \quad pp \to Zg \quad \text{(collinear approximation)}
\end{aligned}
\tag{18}
$$

We generate 1M events for the hard process with SHERPA [35], where the $Z$-boson decays to electrons. For the network we combine the electron and positron momenta to a 4-momentum of the $Z$-boson, so we obtain a Breit–Wigner distribution with $m_{ee} = 66 \dots 116$ GeV instead of an on-shell condition. We then subtract the corresponding Catani-Seymour dipoles [1] for the gluon radiation off each of the incoming quarks, based on 1M events each. The corresponding Feynman diagram is shown in the right panel of Fig. 5. To avoid the soft divergence we require $p_{T,g} > 1$ GeV in the training data, a smaller cutoff would be possible but increase the training time. We apply the same external cutoff to the GANned samples, aligning the phase space boundaries of the training and GANned data sets by hand.

The problem with this specific process is that the Catani-Seymour dipoles describe the full matrix element over a huge part of phase space [36] and the combination of hard matrix element and dipoles is typically tiny and negative. We discuss changing signs in probability distributions in Sec. 2.3. In addition, the one distribution a GAN can never generate is a probability distribution compatible with zero everywhere. In this case the GAN would either over-fit statistical fluctuations or become unstable. This is why in our toy application we shift the Catani-Seymour dipole by a constant such that the cancellation of the divergent matrix element still works, but the combined result integrated over phase space remains finite.

Note that the kinematics of our subtraction terms are not the same as in fixed-order calculations, instead it is similar to the mapping in the modified subtraction method MC@NLO [37]. In this case the global efficiency of event generators at NLO accuracy is dominated by the efficiency of computing the subtracted real-emission corrections, which presents a major challenge for event simulation at the HL-LHC [38, 39].

The hyper-parameters have to be modified with respect to the background subtraction example, due to the large cancellations in the low energy regime. Now, the discriminator and generator networks consist of 8 layers with 256 and 512 units per layer, respectively. In

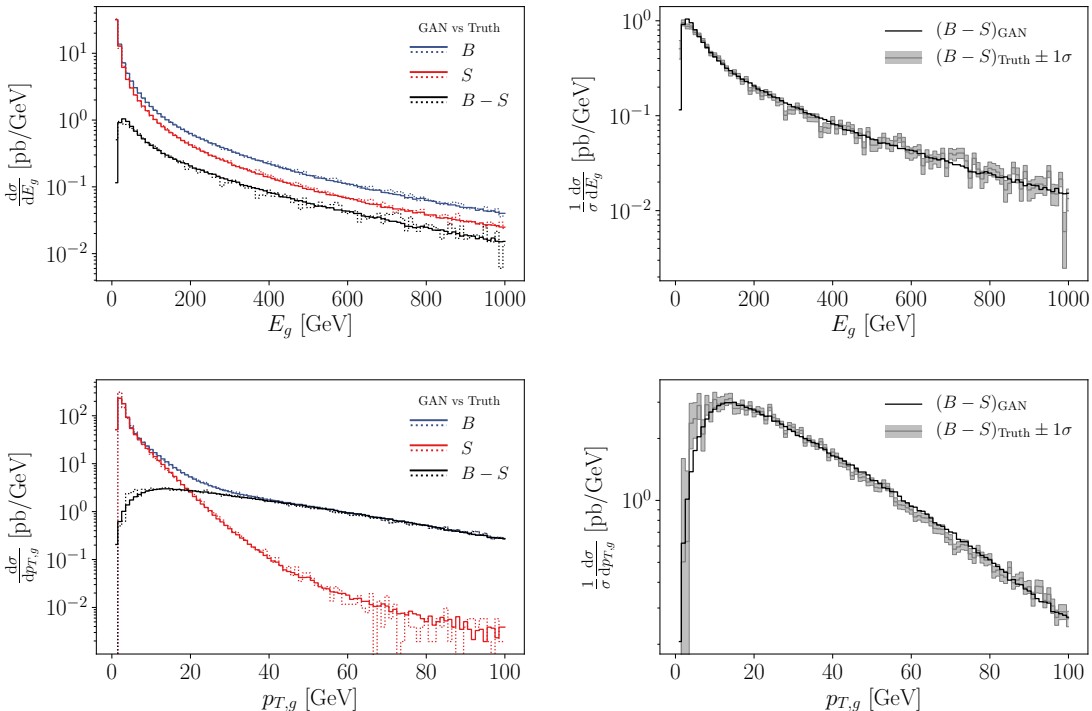

Figure 7: Left: Generated (solid) and true (dashed) $Zg$ events at the LHC for the two input distributions and the subtracted output. Right: distribution of the subtracted events, true and generated, including the error envelope propagated from the input statistics.

the generator we alternate LeakyRelu and tanh activation functions. We achieve the best and most stable results choosing $\lambda_{D_i} = 10^{-3}$ with a batch size of 1024 events. We train for 60000 epochs, where each epoch consists of 5 iterations in which the discriminator gets updated twice as much as the generator. In this example, the discriminator gets the events in the $\{E, p_T, \eta, \phi\}$ representation, which is better suited to resolve the $p_T$ distribution. The other hyper-parameters are kept the same as in the background subtraction.

We show the results from the collinear subtraction in Fig. 7. The GAN perfectly reconstructs the cancellation in the energy spectrum and the transverse momentum of the emitted gluon. The left panel shows the distribution of the real ($B$) and dipole ($S$) contributions to the process and their difference ($B-S$). With the logarithmic axis we see that the GAN smoothly interpolates over the entire energy range. For small gluon energies and momenta the GAN reproduces the rate increase towards the (enforced) phase space boundary, including the finite value of the subtracted combination $B-S$. Also in the high energy region, which suffers from low statistics, the GAN nicely matches the truth distributions. In the right panel we show the subtracted curve as always including the error envelope of the input data.

As before, we only use the established NLO dipole as a simple structure to illustrate the features of our subtraction GAN. Proper applications could be the more complicated subtraction terms beyond NLO or the subtraction of on-shell resonances [8]. The latter would combine aspects discussed in Sec. 3.1 and Sec. 3.2 and allow for a fully inclusive study of the kinematics in the off-shell process, without having to actually do a subtraction and deciding if a given event is more likely to be part of the on-shell or off-shell sample.

# 4 Outlook

We have shown how to generate events representing the difference between two input distributions with a GAN. As a toy example we used events representing a 1-dimensional probability distribution. Because the GAN interpolates the input while learning the difference between the two distributions, it circumvents the statistical limitations of large cancellations. We have found that the GAN-subtracted events lead to a very stable phase space coverage and beat the statistical limitations of the input sample over the entire phase space.

For a slightly more realistic setup we have GANned background subtraction and collinear dipole subtraction for Drell–Yan production at the LHC. In the first case the network learned on-shell final state momenta to subtract the photon-induced continuum from the full $e^+e^-$ production. It could serve as a test case for a background subtraction for four-body decays, such that the GANned signal events reflect the kinematic correlations of the actual signal events hidden in the background.

In the second case we combined the hard matrix element with modified Catani-Seymour dipoles for gluon emission into a stable finite prediction of the real emission process. We are aware of the fact that our toy examples are not more than an illustration of what a subtraction GAN can achieve. However, we have shown how to use a GAN to manipulate event samples avoiding binning (at least in particle physics) and we hope that some of the people who do LHC event simulations for a living will find this technique useful.*

# Acknowledgments

We would like to thank Stefan Höche for extremely useful discussions and for helping us out with specially made SHERPA events on really short notice. In addition, we would like to thank Olivier Mattelaer for his incredibly friendly help with MADGRAPH5. TP would like to thank Alexander Grohsjean for some very helpful discussions on possible applications of subtraction GANs. Finally, we would like to thank Kirill Melnikov for inspiring this project and Gregor Kasieczka for his continuous input on machine learning and GANs. RW acknowledges support by the IMPRS for *Precision Tests of Fundamental Symmetries*. The research of AB and TP is supported by the Deutsche Forschungsgemeinschaft (DFG, German Research Foundation) under grant 396021762 — TRR 257 *Particle Physics Phenomenology after the Higgs Discovery*.

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
