# Peer review of "How to GAN Event Subtraction"

_SciPost Physics, doi:SciPost Phys. Core 3, 009 (2020)_

## Round 2 · Referee Report · Anonymous (Referee 1) · 2020-3-7

Report

A machine-learning method is proposed to construct new event samples that follow a distribution obtained by summing or subtracting given input samples. The method is based on the use of generative adversarial networks (GANs).

The GAN architecture takes two (or more) sets of input events following given distributions and trains a generator whose output follows a distribution corresponding to a linear combination of the given inputs. Typical cases correspond to the sum or subtraction of distributions.

Simple one-dimensional toy examples are considered to explain the general GAN architecture and to test its applicability. In these cases it is shown that the GAN approach can correctly reproduce the subtracted distribution, within the 1-sigma error band derived by a binned analysis.

Two examples with actual LHC montecarlo simulations are also considered. The first one is the subtraction of the photon continuum from the p p -> e+ e- process at the LHC. The second one is the subtraction of the collinear radiation part from the p p -> Z g process. In both cases the GAN method seems able to perform the wanted subtraction.

There are a couple of points that are not fully discussed in the paper, but are crucial to understand the usefulness of the proposed GAN method.

1) The first point regards the reconstruction error associated to the GAN approach. In the toy examples it is shown that the GSN approach is able to reproduce the target distribution within the 1-sigma error band obtained from a binned analysis. This result, however, might be strongly influenced by the hyper-parameters, i.e. the neural network structure, the training parameters and the training algorithm. In all the examples the hyper-parameters are carefully adjusted to obtain a good performance.

It is not at all clear and not discussed in the text, how the reconstruction error is influenced by those choices. In a more realistic set-up, in which one can not compute the error from a binned analysis, one would not know a priori the error associated to the GAN reconstruction. This might be an issue if one wants to use these techniques for physics analyses, in which all the sources of systematic errors should be carefully estimated and taken into account.

Is there a way to get such an estimate for the GAN approach?

Notice that, at the end of the Outlook section, there is the sentence “we have shown how to use a GAN to manipulate event samples avoiding binning”. Therefore it seems clear that this method is proposed as an alternative to binning. As such, a proper treatment of errors would be needed.

2) The examples discussed in the paper do not seem to be particularly useful from the LHC point of view (“We are aware of the fact that our toy examples are not more than an illustration of what a subtraction GAN can achieve”, taken from the Outlook section). Although the hope that the method is used for actual LHC analyses is expressed (“we hope that some of the people who do LHC event simulations will find this technique useful”), there is no mention to possible “useful” applications. Do the Authors know any example of “useful” application of the GAN technique?

  • validity: high
  • significance: good
  • originality: high
  • clarity: good
  • formatting: -
  • grammar: -

Author:  Ramon Winterhalder  on 2020-05-12  [id 821]

(in reply to Report 1 on 2020-03-07)

1) It is not at all clear and not discussed in the text, how the reconstruction error is influenced by those choices. In a more realistic set-up, in which one can not compute the error from a binned analysis, one would not know a priori the error associated to the GAN reconstruction. This might be an issue if one wants to use these techniques for physics analyses, in which all the sources of systematic errors should be carefully estimated and taken into account. Is there a way to get such an estimate for the GAN approach?

-> What should we say - we are not aware of a serious study of uncertainties in generative networks, but we are working on it. As a matter of fact, we are starting a more serious collaboration on this crucial LHC question with local ML experts...

Notice that, at the end of the Outlook section, there is the sentence “we have shown how to use a GAN to manipulate event samples avoiding binning”. Therefore it seems clear that this method is proposed as an alternative to binning. As such, a proper treatment of errors would be needed.

-> We completely agree, this paper is really meant as another motivation for the major effort of studying errors in GAN output. We added a comment along this line to Sec.2.

2) The examples discussed in the paper do not seem to be particularly useful from the LHC point of view (“We are aware of the fact that our toy examples are not more than an illustration of what a subtraction GAN can achieve”, taken from the Outlook section). Although the hope that the method is used for actual LHC analyses is expressed (“we hope that some of the people who do LHC event simulations will find this technique useful”), there is no mention to possible “useful” applications. Do the Authors know any example of “useful” application of the GAN technique?

-> We changed the introduction, the respective sections, and the outlook accordingly. Now there should be a clearer picture of where such an event subtraction might come in handy.

---

## Round 2 · Referee Report · Ayan Paul (Referee 2) · 2020-3-18

Report

I have provided a detailed report in the attached file. I do not recommend this paper for publication.

Attachment

  • validity: ok
  • significance: ok
  • originality: good
  • clarity: low
  • formatting: good
  • grammar: good

Author:  Ramon Winterhalder  on 2020-05-12  [id 822]

(in reply to Report 2 by Ayan Paul on 2020-03-18)

-> We fear there is a misunderstanding in our problem statement - our goal is to construct a network that can generate events according to the difference of two probability distributions. The referee's network does an excellent job in constructing the distribution corresponding to the difference of two event samples, but it cannot be extended to generate statistically independent events.

  1. I do not think Ref.[11] uses a generative network. They use a DNN and show that they perform better than Ref.[12] which uses a GAN. The authors can maybe take a deeper look into thesepapers.

-> Thank you for pointing this out, we wanted to cite Ref.[11] alongside with the generative phase-space studies, took it out now.

  1. The authors do not provide the code that they use or any details about it or what framework they used (PyTorch/Sci-Kit Learn/ TensorFlow etc.). The authors also do not provide the data they used for the training. While this is not necessary, it is useful to have it if someonewants to reproduce their results. I would suggest the authors provide all these details (possiblyin a public repository) and also an example code since the work is primarily computational.

-> We added a footnote clarifying that our code and out test data are available upon request. We also added details on our software.

  1. The authors do not make explicit the training times and the hardware used for training the GANs. This is useful to benchmark it against other regression methods.

-> As mentioned above we have doubts that this helps in comparing with regression networks, given that we do not actually do a regression :) In any case, we find that quoting such numbers are not helpful in a field with collaborative spirit, but we have a track record of happily participating in proper comparison studies.

  1. The authors do not describe how they get the error-bars in the left panels of Fig.2, Fig.3, etc. Are they from Eq.(1)?

-> They are, and we clarified this in the text.

---

## Round 2 · Referee Report · Anonymous (Referee 3) · 2020-3-22

Strengths

The application of GAN techniques discussed in the manuscript is new, as far as I can tell, and it is quite clearly explained in the manuscript.

Weaknesses

The actual domain of applicability of the method to concrete collider problems is not clear.

Report

The manuscript reports on the idea of using GANs to perform the following operation. Using two sets of events (say, "B" and "S", possibly generated by two Monte Carlo generators), one can train a GAN to produce a generator of events that follow the difference between the distributions of the two original sets (i.e., "B-S"). The method is applied to toy problems, with the aim of outlining potential applications to collider phenomenology.

This application of GAN techniques is new, as far as I can tell, and it is quite clearly explained in the manuscript. The manuscript however leaves a number of open questions concerning the actual domain of applicability of the method to concrete collider problems.

In light of this, I believe that the manuscript is not ready for publication in the current form. If the authors can address the points reported below, and clarify the potential of the method to address concrete problems (and convincingly support its competitive advantages), one could address other less crucial aspects related with the presentation of the GAN algorithm and the comparison with other strategies that might be employed for the same task, which should perhaps be slightly extended.

Requested changes

The following points should be addressed before the manuscript can be considered for publication:

1- It is unclear what happens if "B-S" does not have definite sign. Namely if the event density distribution P_B(x) is larger than P_S(x) in some region of "x", and smaller than P_S(x) in some other region. In this case, neither P_S-P_B nor P_B-P_S are densities, and the problem seems ill-defined. Since this can happen in potential applications (see below), one should ask what the GAN would return if trained on a problem of this type, and if the method would at least allow one to recognise that there is an issue or it would instead produce wrong results.

On page 11 it is mentioned that the sign of "B-S" is not a problem because one can always learn "S-B" instead of "B-S". However this seems to assume that "B-S" has fixed sign on the entire feature (x) space, so this comment is not sufficient to address the question above.

2- The first class of applications mentioned in the manuscript are referred to as "background subtraction". However I could not find a discussion of what this should be concretely useful for. The example worked out in the manuscript (photon background subtracted from Drell-Yan, in section 3.1) does not shed light on this aspect because it is not clear why one might want to perform such subtraction.

Maybe the method is supposed to help for problems such as extracting the new physics contribution from a simulation containing also the standard model, for instance in cases where the new physics effect is small and the approach based on bins becomes computationally unfeasible. If this is the case, it should be clearly stated in the manuscript. However one should also take into account that performing a subtraction would be needed only if simulating the new physics contribution separately is not feasible. This is the case in the presence of quantum-mechanical interference between SM and new physics. However in the presence of interference, "B-S" does not have definite sign in general. So the feasibility and the usefulness of the approach in this domain depends on point "1)".

3- The second class of applications are "subtraction" (see section 3.2). Also in this case, the final goal is not clearly stated in the paper. A short paragraph at the end of page 11 alludes to the fact that this could help MC@NLO event generation. If this is the case, it should be clearly stated and extensively explained. Also, it is found in section 3.2 that the required task of subtracting the collinear contribution cannot be accomplished because the method cannot deal with "B-S" distributions that are very small. Would this prevent the method to work, eventually?

  • validity: ok
  • significance: ok
  • originality: high
  • clarity: ok
  • formatting: -
  • grammar: -

Author:  Ramon Winterhalder  on 2020-05-12  [id 823]

(in reply to Report 3 on 2020-03-22)

1- It is unclear what happens if "B-S" does not have definite sign. Namely if the event density distribution P_B(x) is larger than P_S(x) in some region of "x", and smaller than P_S(x) in some other region. In this case, neither P_S-P_B nor P_B-P_S are densities, and the problem seems ill-defined. Since this can happen in potential applications (see below), one should ask what the GAN would return if trained on a problem of this type, and if the method would at least allow one to recognise that there is an issue or it would instead produce wrong results.

-> We expanded the discussion of signs and the zero function in Sec.2.3. As a matter of fact, our CS-like example already has such a sign problem which we solve with an off-set.

2- The first class of applications mentioned in the manuscript are referred to as "background subtraction". However I could not find a discussion of what this should be concretely useful for. The example worked out in the manuscript (photon background subtracted from Drell-Yan, in section 3.1) does not shed light on this aspect because it is not clear why one might want to perform such subtraction.

Maybe the method is supposed to help for problems such as extracting the new physics contribution from a simulation containing also the standard model, for instance in cases where the new physics effect is small and the approach based on bins becomes computationally unfeasible. If this is the case, it should be clearly stated in the manuscript. However one should also take into account that performing a subtraction would be needed only if simulating the new physics contribution separately is not feasible. This is the case in the presence of quantum-mechanical interference between SM and new physics. However in the presence of interference, "B-S" does not have definite sign in general. So the feasibility and the usefulness of the approach in this domain depends on point "1)".

-> Again, we admit that we only work with a toy model. We now add a brief discussion of an appropriate problem, namely the kinematics of a GANned 4-body-decay signal from signal-plus-background and background samples.

3- The second class of applications are "subtraction" (see section 3.2). Also in this case, the final goal is not clearly stated in the paper. A short paragraph at the end of page 11 alludes to the fact that this could help MC@NLO event generation. If this is the case, it should be clearly stated and extensively explained. Also, it is found in section 3.2 that the required task of subtracting the collinear contribution cannot be accomplished because the method cannot deal with "B-S" distributions that are very small. Would this prevent the method to work, eventually?

-> We added some more discussion, including the subtraction of on-shell events as a combination of the two examples. However, we admit that we are not MC authors with a clear vision where exactly such a tool would enter which MC code. We also improved the numerics in Sec.3.2 to show that given some more optimization and enough training time we do not expect precision to be an immediate show stopper.

-> Altogether, we would like to thank the three referees and everyone who has discussed with us since the first version of the paper came out. We have changed the paper in many places, including abstract, introduction, physics discussions, and outlook. This is why we are confident that the current version is significantly improved over the original draft and hope that SciPost agrees with that judgement.

---

## Round 3 · Referee Report · Ayan Paul (Referee 2) · 2020-5-28

Report

--> We fear there is a misunderstanding in our problem statement – our goal is to construct a network that can generate events according to the difference of two probability distributions. The referee's network does an excellent job in constructing the distribution corresponding to the difference of two event samples, but it cannot be extended to generate statistically independent events.

Let me clarify that there is no misunderstanding on my part. Also, let me try to explain why. Generating a statistically independent sample from distributions constitute of two parts:
1. A regression/interpolation of the underlying distribution from data points generated by simulation or experiments (real data).
2. A sampling from the underlying distribution resulting from 1.

The part described in 1. is the more complex part and its complexity depends on the complexity of the underlying distribution that needs to be fit to. The following options are available. The list is neither comprehensive nor are the elements mutually exclusive:
a. Simple Interpolation, linear, spline, Steffen, Akima depending on what features one wants from the interpolation
b. Regression using a polynomial
c. Regression using a basis function set (akin to neural networks)
d. Regression using binary splitting like a decision tree and its variants
e. Regression using a support vector machine with a non-linear kernel
f. Deep Neural Networks (Dense, Convolutional, Recurrent etc.)
g. Gaussian process regression
h. Markov Chain Monte Carlo (with something like Metropolis-Hastings algorithm)
i. GAN

GANs are literally at the end of the evolutionary phylogenic tree of regression/classification methods that are generative. They were built to sample complex multidimensional distributions which can be highly multimodal and have translational invariance (panda on the left is the same as a panda on the right of a picture). This does not mean a GAN cannot be used for the regression of two simple one-dimensional functions that are at best uni- or bi-modal. But it is an overkill.
It would be alright to use a GAN for simple regression, but the hardware overhead and the time consumed for its training does not allow it to have any advantage over the other methods. In fact, both Gaussian processes and MCMC can be built in a generative manner very easily and can deliver the same precision as a GAN (sometimes better) for a simple distribution and perform much faster.
Often it is argued that one needs to know the underlying distribution to use an MCMC. But that is true for any of the methods since one needs to always assume a basis function set in any regression (called activation functions in the cases of neural networks). MCMC is also capable of handling highly multivariate distributions but does not perform as well with highly multimodal distributions. Hence GANS outperform when things like jet clustering or Cherenkov rings need to be generated after learning from a set of realistic samples.

The part in 2. Is the simpler part, follows similar strategies for 1a. – 1f. and involves only a few lines of codes and is typically very fast in computational time. There is about a century worth of studies in Statistics that already exists on how this can be done and several existing packages allow the completion of this part in a few lines of code. Typically this part is not time-consuming at all. The generation of the sample is automatically accounted for 1g. – 1f.

The authors argued that GANs reduce errors of subtraction. It is not a property of the GAN. Rather, if one does a regression of the underlying distribution using any reasonable method listed above one will get a reduced error compared to bib-by-bin subtraction. This is because one is using a large number of bins to fit a lower-dimensional function and sampling from it. It’s a statistical truth. If, however, one takes into account the bin error and the bin correlation the errors will be inflated since this is the correct way to do it anyway. This is why a comprehensive understanding of error propagation in GANs is necessary as one of the other referees pointed out. In fact, https://arxiv.org/abs/2002.06307 which cites this work clearly cautions against the use of GAN as is done in this work.

--> As mentioned above we have doubts that this helps in comparing with regression networks, given that we do not actually do a regression:) In any case, we find that quoting such numbers are not helpful in a field with collaborative spirit, but we have a track record of happily participating in proper comparison studies.

GANs are after all an adversary (discriminator) learning a regression from a training set while co-training a generator to generate events. The authors can take a look at the original GAN paper: https://arxiv.org/abs/1406.2661. Equation 1 simply is a sum of objective functions for the discriminator and generator and algorithm 1 gives the flow for the gradient descent used for training. The discriminator is performing a regression (or classification) while the generator generates samples which the discriminator accepts or rejects. As the training goes on, the discriminator gets better at telling apart fake from real samples and the generator gets better at generating fake samples that look real. Whether one wants to do a regression or classification determines the objective function and network construction.

I reiterate my decision more clearly. It is a nice exercise to show that GANs can be used to subtract samples from separate distributions and combine them. However, it is not sufficient to be a stand-alone publication especially because the same can be done with off-the-shelf packages with very few lines of coding and there is no innovation that the authors made to make this work from the Statistics/Numerical methods standpoint. Its is a straightforward application of existing neural network methods. From the Physics standpoint, the subtraction of events can be done a lot more efficiently by simple statistical methods that are well established. If the authors were to consider far more complex distributions that cannot be generated by simpler methods or if they showed GANs perform much better than other methods of sampling from numerical distributions, it would merit consideration. I cannot recommend this paper for publication even after the modifications.

---

## Round 3 · Referee Report · Anonymous (Referee 1) · 2020-6-8

Strengths

Novelty of the proposed method.

Weaknesses

No clear indication of possible applications of the proposed method. Some crucial aspects of the method (estimation of errors) are not under control.

Report

Dear Editor,

the answers to the points I discussed in the previous report do not seem satisfactory.

1) The Authors confirm that the method they propose is an alternative to binning. However they have no idea of how even a rough estimate of the errors can be obtained. If this is the case the method is clearly useless and there is no hope to adopt it for a quantitative collider analysis. Showing through a plot that in two toy examples the errors seem under control does not give any proof of its applicability to contexts where a comparison with alternative methods can not be performed.

2) The changes in the introduction and in the outlook are a simple rewriting of the old text. They do not offer any hint of possible applications of the proposed method to actual collider analyses.

I think that the manuscript in the present form can not be accepted for publication.

  • validity: -
  • significance: -
  • originality: -
  • clarity: -
  • formatting: -
  • grammar: -

Author:  Tilman Plehn  on 2020-08-31  [id 939]

(in reply to Report 2 on 2020-06-08)
Category:
remark

We are not really sure what the referee is complaining about. In the current version we state clearly where our method can be useful, both on the simulation side and on the analysis side. We are in contact with interested experimentalists, who understood that and are looking into the analysis benefits in more detail.

We are also confused by the `useless' statement, because simple error propagation shows that our method will beat binning approaches, the question is by only how much.

Concerning the errors - of course there are ways to estimate uncertainties the same way as it is done in standard analyses, based on Monte Carlo or based on control regions. Our statement only concerns the error from the network modelling. This is similar to multivariate classifiers used all over ATLAS and CMS, where model uncertainties are always evaluated indirectly. As another example we would like to mention the sPlot approach and the assumed de-correlation, which has to be proven after the fact as well.

---

## Round 3 · Referee Report · Anonymous (Referee 4) · 2020-9-15

Strengths

1) The manuscript shows for the first time how a GAN can be trained to generate events that follow probability distributions functions that are the subtraction or sum of other distributions, with the output distribution being much smaller than the others.

2) The discussion on the GAN training and how all the results are obtained is clear and concise.

Weaknesses

1) It is not clear what the usefulness of the method shown would be in the domain of event generation for colliders.

2) It is not clearly stated what the benefits of using a GAN instead of a standard Monte Carlo generator are.

3) The error of the GAN output is not estimated, but this can be overlooked due to the novelty of the method.

Report

The manuscript shows how GAN models can generate events following probability distribution functions (PDF) that are the sum or subtraction of other PDFs, with the output PDF being much smaller than the others.

Nonetheless, there is no concrete discussion on what problems are being addressed by the method presented, nor why one would give up on Monte Carlo generators and motivate GANs.
Moreover, there is no mention on why GANs are better solutions than other Machine Learning (ML) approaches to the problems being addressed, for instance fitting the subtracted distribution and generating events from it.

Currently, there is no way to estimate the error of a GAN output, making this approach unusable for the time being. Even if this is the case, the paper is of good quality and if the changes requested below are addressed I would gladly recommend it for publication.

Requested changes

1) In the current manuscript, there is not a clear physics case stated where one would require to generate events that follow a probability distribution function (PDF) which is the result of the addition or subtraction of two or more other PDFs that can not be addressed by standard MC event generators.

  • Adding a concise and general physics case for the need of generating such events with a GAN in the context of collider physics is a must in order to understand the usefulness of the proposed method. It should be clearly stated what are the advantages of using a GAN to generate subtracted events with respect to a standard MC event generator and also the cases where the generation of such events is necessary.

-What is the problem that you are trying to address? And then, why is the use of GAN preferred to standard MC event generators?

  • For instance in Section 3.2 it is not clearly stated why one would want to use a GAN instead of a MC generator to generate the events. What is the concrete problem that you are trying to address in this section?

2) It would be very useful and informative if a small discussion on the advantages of using GANs, with respect to other ML approaches, would be given. For instance, why are GANs better to generate subtracted events as opposed to fitting the subtracted distribution and generating events from it.

3) In some regions of Fig. 1 (right) and Fig. 7 (bottom-right), the errors of the GAN distribution are biased with respect to the true distribution (they are either always above or below in certain regions).

If one would integrate the a GAN distribution with regions that are biased, for instance in Fig. 7 (bottom-right) from 0 to 20 GeV or from 60 to 80 (or in Fig. 1 from 0 to 25), then the mean value of the bin resulting from integrating out the GAN values, could in principle be outside the statistical error given by the truth distribution in that bin.

  • Do these biases make the GAN generation of events much less reliable when binning the events?
  • How would you address this problem?

4) It would be useful, in order to compare with standard MC event generators and other ML approaches, if you could report the timings for training and inference, together with the hardware used, at least for sections 3.1 and 3.2.

  • validity: ok
  • significance: good
  • originality: good
  • clarity: good
  • formatting: excellent
  • grammar: -

Author:  Ramon Winterhalder  on 2020-10-16  [id 1009]

(in reply to Report 3 on 2020-09-15)

We are thankful for the referee's feedback and his comments. As some questions and remarks have been raised, we would like to answer them in the following:

  • First of all, we note the referee's concern about the actual requirement for such a technique in a real world physics case. In the paper we give two examples. First we point out in section 3.2 that this kind of subtraction of individually generated samples is required in the modified subtraction method introduced in the MC@NLO approach (arXiv:hep-ph/0204244) which is commonly used to match NLO calculations with parton shower simulations. We now added a more detailed explanation about the MC@NLO approach to the paper which should make the necessity for our technique more obvious.

  • Further, the problem of event subtraction in a consistent high-dimensional way is a highly-relevant question in LHC simulations and in LHC analyses. As illustrated with our simple signal-background subtraction we wanted to point out the relevance in more sophisticated applications like 4-body decays in which a bin-wise subtraction would lose many kinematic properties. Since our method operates on distributions represented by samples, it can be applied to data. We also would like to mention the sPlot approach, which is now cited in the paper in section 3.1.

  • In this paper we did not address the general question why to use GANs for event generation. However, as we have shortly pointed out in our introduction, the application and need for GANs to supplement standard MC generators have already been discussed in detail, for instance in Ref. 1901.05282, 1903.02433, 1903.02556, 1907.03764, 1912.02748, 2001.11103 etc., which we have cited accordingly. We now refer more explicitly to those papers.

  • In the application of subtraction, the benefits of using a GAN instead of first fitting the subtraction distribution and then sampling from it are twofold. First, in many cases the subtraction distribution is not directly available and hence a fitting not feasible. Further, fitting the distribution and then using sampling from it involves two different steps which can be accommodated into one using a GAN.

  • The observed biases in the GAN output are indeed a current problem and show up in almost all carefully performed studies of GANs for event generation. As it was pointed out in for instance Ref. 1907.03764 these biases and hence systematic effects are mostly originating from low statistics regions in the training data. However, they are also linked to uncertainties in the training of GANs. The latter one is the biggest open question today. It is generally not yet clear how systematic effects and uncertainties can be estimated when using GANs and many groups are currently working on a solution.

  • For training we employed a NVIDIA GPU and the training time was at the order of ~minutes and the inference/evaluation of the trained network is performed in ~ms.

---

## Round 3 · Author Response

Referee 1:

1) It is not at all clear and not discussed in the text, how the reconstruction error is influenced by those choices. In a more realistic set-up, in which one can not compute the error from a binned analysis, one would not know a priori the error associated to the GAN reconstruction. This might be an issue if one wants to use these techniques for physics analyses, in which all the sources of systematic errors should be carefully estimated and taken into account. Is there a way to get such an estimate for the GAN approach?

-> What should we say - we are not aware of a serious study of uncertainties in generative networks, but we are working on it. As a matter of fact, we are starting a more serious collaboration on this crucial LHC question with local ML experts...

Notice that, at the end of the Outlook section, there is the sentence “we have shown how to use a GAN to manipulate event samples avoiding binning”. Therefore it seems clear that this method is proposed as an alternative to binning. As such, a proper treatment of errors would be needed.

-> We completely agree, this paper is really meant as another motivation for the major effort of studying errors in GAN output. We added a comment along this line to Sec.2.

2) The examples discussed in the paper do not seem to be particularly useful from the LHC point of view (“We are aware of the fact that our toy examples are not more than an illustration of what a subtraction GAN can achieve”, taken from the Outlook section). Although the hope that the method is used for actual LHC analyses is expressed (“we hope that some of the people who do LHC event simulations will find this technique useful”), there is no mention to possible “useful” applications. Do the Authors know any example of “useful” application of the GAN technique?

-> We changed the introduction, the respective sections, and the outlook accordingly. Now there should be a clearer picture of where such an event subtraction might come in handy.

Referee 2:

-> We fear there is a misunderstanding in our problem statement - our goal is to construct a network that can generate events according to the difference of two probability distributions. The referee's network does an excellent job in constructing the distribution corresponding to the difference of two event samples, but it cannot be extended to generate statistically independent events.

  1. I do not think Ref.[11] uses a generative network. They use a DNN and show that they perform better than Ref.[12] which uses a GAN. The authors can maybe take a deeper look into thesepapers.

-> Thank you for pointing this out, we wanted to cite Ref.[11] alongside with the generative phase-space studies, took it out now.

  1. The authors do not provide the code that they use or any details about it or what framework they used (PyTorch/Sci-Kit Learn/ TensorFlow etc.). The authors also do not provide the data they used for the training. While this is not necessary, it is useful to have it if someonewants to reproduce their results. I would suggest the authors provide all these details (possiblyin a public repository) and also an example code since the work is primarily computational.

-> We added a footnote clarifying that our code and out test data are available upon request. We also added details on our software.

  1. The authors do not make explicit the training times and the hardware used for training the GANs. This is useful to benchmark it against other regression methods.

-> As mentioned above we have doubts that this helps in comparing with regression networks, given that we do not actually do a regression :) In any case, we find that quoting such numbers are not helpful in a field with collaborative spirit, but we have a track record of happily participating in proper comparison studies.

  1. The authors do not describe how they get the error-bars in the left panels of Fig.2, Fig.3, etc. Are they from Eq.(1)?

-> They are, and we clarified this in the text.

Referee 3:

1- It is unclear what happens if "B-S" does not have definite sign. Namely if the event density distribution P_B(x) is larger than P_S(x) in some region of "x", and smaller than P_S(x) in some other region. In this case, neither P_S-P_B nor P_B-P_S are densities, and the problem seems ill-defined. Since this can happen in potential applications (see below), one should ask what the GAN would return if trained on a problem of this type, and if the method would at least allow one to recognise that there is an issue or it would instead produce wrong results.

-> We expanded the discussion of signs and the zero function in Sec.2.3. As a matter of fact, our CS-like example already has such a sign problem which we solve with an off-set.

2- The first class of applications mentioned in the manuscript are referred to as "background subtraction". However I could not find a discussion of what this should be concretely useful for. The example worked out in the manuscript (photon background subtracted from Drell-Yan, in section 3.1) does not shed light on this aspect because it is not clear why one might want to perform such subtraction.

Maybe the method is supposed to help for problems such as extracting the new physics contribution from a simulation containing also the standard model, for instance in cases where the new physics effect is small and the approach based on bins becomes computationally unfeasible. If this is the case, it should be clearly stated in the manuscript. However one should also take into account that performing a subtraction would be needed only if simulating the new physics contribution separately is not feasible. This is the case in the presence of quantum-mechanical interference between SM and new physics. However in the presence of interference, "B-S" does not have definite sign in general. So the feasibility and the usefulness of the approach in this domain depends on point "1)".

-> Again, we admit that we only work with a toy model. We now add a brief discussion of an appropriate problem, namely the kinematics of a GANned 4-body-decay signal from signal-plus-background and background samples.

3- The second class of applications are "subtraction" (see section 3.2). Also in this case, the final goal is not clearly stated in the paper. A short paragraph at the end of page 11 alludes to the fact that this could help MC@NLO event generation. If this is the case, it should be clearly stated and extensively explained. Also, it is found in section 3.2 that the required task of subtracting the collinear contribution cannot be accomplished because the method cannot deal with "B-S" distributions that are very small. Would this prevent the method to work, eventually?

-> We added some more discussion, including the subtraction of on-shell events as a combination of the two examples. However, we admit that we are not MC authors with a clear vision where exactly such a tool would enter which MC code. We also improved the numerics in Sec.3.2 to show that given some more optimization and enough training time we do not expect precision to be an immediate show stopper.

-> Altogether, we would like to thank the three referees and everyone who has discussed with us since the first version of the paper came out. We have changed the paper in many places, including abstract, introduction, physics discussions, and outlook. This is why we are confident that the current version is significantly improved over the original draft and hope that SciPost agrees with that judgement.

---

## Editorial Decision

published